# Design of Optical-Wireless IR-UWBoF Systems with Spectral Line Suppression Capabilities

**Aldo-Eleazar Perez-Ramos** [1], **Salvador Villarreal-Reyes** [2,*], **Alejandro Galaviz-Mosqueda** [3] **and Catherine Lepers** [4]

1 Department of Electronic Engineering, TecNM—Instituto Tecnologico de Oaxaca, Oaxaca 68030, Oaxaca, Mexico
2 Department of Electronics and Telecommunications, CICESE Research Center, Ensenada 22860, Baja California, Mexico
3 Monterrey Unit, CICESE Research Center, Apodaca 66629, Nuevo Leon, Mexico
4 Telecom SudParis, Institut Polytechnique de Paris, 91764 Palaiseau, France
* Correspondence: svillar@cicese.edu.mx

**Abstract:** Impulse-Radio Ultra-Wide Band (IR-UWB) over Fiber (IR-UWBoF) has been proposed to interconnect IR-UWB-based deployments separated by hundreds of meters or even kilometers. IR-UWB transmissions must comply with spectral masks provided by radio spectrum regulatory agencies. The maximum transmit power of an IR-UWB signal is adversely affected by the presence of spectral lines in its Power Spectral Density (PSD). Thus, it is desirable that the PSD of signals generated by IR-UWBoF systems does not show spectral lines. Previous works have shown the feasibility of deploying of optical-wireless IR-UWBoF systems. However, most of these proposals report PSDs showing spectral lines. To the best of our knowledge, spectral line suppression has not been previously studied for optical-wireless IR-UWBoF systems. This work shows the design and implementation of optical-wireless IR-UWBoF systems generating signals with Spectral Line-Free (SLF) PSDs. The proposal considers the use the use of a specifically designed convolutional code combined with Binary Phase Shift Keying (BPSK) or Quaternary Biorthogonal Pulse Position Modulation (Q-BOPPM) to provide a SLF PSD in IR-UWBoF systems. A testbed consisting of 30 km of single-mode optical fiber (SMF) concatenated to a 20 cm wireless link was physically implemented. The results show that a SLF PSD is achieved for both the optical and the wireless transmissions, even when the binary data source feeding the system is not perfectly random.

**Keywords:** UWB systems; IR-UWB over fiber systems; power spectral density; spectral line suppression; IM/DD method; convolutional coding

## 1. Introduction

Recent advances in microelectronics, sensors, protocols and wireless technologies have led to new ways of sharing information between humans and things, either locally or globally, through the Internet, [1–6]. At present, wireless services such as ultra-high-definition video and immersive media, high-speed massive file transfers, and telemonitoring systems are becoming ubiquitous, [7,8]. Most of these wireless services use mature technologies operating in the Industrial, Scientific and Medical (ISM) frequency bands. Thus, technologies such as Wi-Fi, Bluetooth and ZigBee have been extensively used to deploy wireless local and personal area networks (WLAN/WPAN) in households, buildings, airports, bus stations, shopping malls, etcetera. Furthermore, the Internet of Things also considers the use of ISM wireless technologies to collect data generated by sensors [9]. Therefore, a rapid ISM frequency band saturation is expected in the near future, e.g., Cisco predicts that more than 500 billion devices will be connected to the Internet by 2030, [10]. Several wireless devices transmitting concurrently over the same radio coverage areas and frequency bands

cause the well-known radio frequency (RF) interference problem, degrading the wireless link performance for all users [11–13].

In recent years, improvements at physical (PHY) and medium access control (MAC) layers of ISM and alternative wireless technologies have been proposed to mitigate the problems caused by the saturation of ISM frequency bands, [14–20]. One of these proposals is Impulse-Radio Ultra-Wideband (IR-UWB), which enables the deployment of robust short-range wireless networks with high node density, low complexity, high tolerance against interference and low power consumption, [14,21–25].

Wireless transmission of IR-UWB signals is regulated by maximum transmit power limits defined by regulatory agencies in the form of spectral masks, [23–25]. The spectral masks aim is to enable harmonious coexistence between UWB systems and previously deployed wireless communication systems. For example, the Federal Communications Commission (FCC) UWB spectral masks define that the power spectral density (PSD) of UWB signals must not exceed a power limit of $-41.3$ dBm/MHz from 3.1. GHz to 10.6 GHz, [22]. In this context, the analysis, estimation, and shaping of the PSD of IR-UWB signals is a topic of major interest for the design of compliant IR-UWB systems.

Typically, IR-UWB systems use short pulses (of the order of nanoseconds) to convey information using On-Off Keying (OOK), Pulse Position Modulation (PPM), Binary Phase-Shift Keying (BPSK), or Biorthogonal PPM (BOPPM), among others, [23,25]. Additionally, IR-UWB signals are commonly designed to have a very low-duty cycle. Because of this characteristic, its low transmit power (as required for spectral mask compliance) and its robustness against interference, the use of IR-UWB technology has been proposed for the deployment of low-power, short-range, multi-hop wireless sensor networks (WSN) with high node densities. Although an UWB-based WSN can be used to collect data generated by sensors deployed in a particular area of interest, solutions such as those reported in [26–31] require information to be exchanged between two or more IR-UWB networks separated by hundreds or thousands of meters. For this reason, several IR-UWB-over-fiber (IR-UWBoF) systems have been previously proposed in the literature to interconnect high- and low-data rate UWB deployments, [32–52] (also see Table II in [41]).

As previously mentioned, one of the main constraints in the design of UWB systems is compliance with spectral masks. In this sense, previous studies addressing the design of IR-UWBoF systems usually report measured and/or analytical PSDs and compare them with spectral masks such as the one defined by the FCC [32–52]. Notoriously, most of the PSDs reported in these works exhibit spectral lines. The problem with the presence of spectral lines is that, in order to comply with the spectral mask limits, a UWB signal whose PSD shows spectral lines will have to be transmitted with less power than a UWB signal with a spectral line-free (SLF) PSD. Therefore, in order to achieve maximum transmit power in the wireless link while fulfilling the spectral mask requirements, it is desirable to design IR-UWBoF systems with a SLF PSD.

Theoretically, if a BPSK IR-UWB system transmits perfectly random binary data with uniform distribution, the PSD of the transmitted signal will not show any spectral lines, [25]. However, spectral lines in the PSD of BPSK IR-UWB signals appear when the binary data stream is not perfectly random (i.e., nonuniformly distributed) [25], which is the most common case in practical sensor network applications. Furthermore, even though the IR-UWBoF systems proposed in [33,47,50,52] use BPSK schemes with very long pseudorandom sequences (i.e., resembling a perfectly random binary data stream with uniform distribution), the experimental PSDs reported in these works show spectral lines. These spectral lines could have been caused by the photonic configuration implemented in the reported setups. However, the spectral line presence should be avoided in order to maximize the transmit power while maintaining compliance with the spectral masks (as previously mentioned). Thus, it is desirable to design BPSK IR-UWBoF systems whose PSD is SLF, even when the data stream consists of nonuniformly distributed bits (i.e., non-perfectly random binary data streams).

Previously in [40], we reported the practical implementation of a SLF IR-UWBoF system. However, the results reported in [40] only covered the effects of the optical link over the PSD. This work extends the design and analysis of spectral line-free IR-UWBoF systems by evaluating a concatenated optical-wireless channel. The proposed system was experimentally evaluated considering a 30 km SMF link connected to a short-range wireless transmission. The practical setup was used to show that a SLF PSD over the wireless link can be achieved when using the BPSK/quaternary BOPPM (Q-BOPPM) IR-UWBoF system proposed in this work, even when the input to the systems consists of a nonuniformly distributed binary data stream. To the best of our knowledge, there is no similar approach reported in the literature for IR-UWBoF systems. To further support this, Table 1 summarizes the main characteristics of previous art dealing with IR-UWBoF systems that report simulated or measured PSDs relevant to this work.

**Table 1.** Previous art of IR-UWB over fiber systems.

| Ref | Type of Study | Data Rate (Gbps) | Optical Channel (km) | Wireless Channel (cm) | IR-UWB Modulation Scheme | Channel Coding | Spectral Line-Free PSD |
|---|---|---|---|---|---|---|---|
| [33] | Analytical Simulation | 1.0/5.0 | SMF (60/120) | 0 | BPSK/OOK | FEC | NO |
| [34] | Analytical Simulation | 2.0 | GI-MMF (1) | 0 | OOK, PPM, PSM | NO | NO |
| [35] | Analytical Experimental | No reported | No reported | 10, 20, 30 | No reported | NO | NO |
| [36] | Simulation | 2.0 | SMF (10) | 0 | OOK | NO | NO |
| [37] | Simulation | 1.0/0.5 | SMF + SOA (160) | 0 | OOK/PPM | NO | NO |
| [38] | Analytical Simulation | 1.25 | SMF (60) | 40 | OOK | NO | NO |
| [39] | Analytical Experimental | 0.004 | SMF (30) | 0 | 16-PPM | CC | NO |
| [40] | Analytical Experimental | 1.0 | SMF (30) | 0 | BPPM, BPSK, Q-BOPPM | CC | YES SLF-CC |
| [42] | Experimental | 1.0 | SMF (23) | 40 | OOK | NO | NO |
| [44] | Experimental | 1.25 | NZDSF (20) | 35 | OOK | NO | NO |
| [45] | Experimental | 2.5 | SMF (59.2) | 0 | OOK | LPDC | NO |
| [46] | Experimental | 2.0 | NZDSF (20) | 800 | OOK | NO | NO |
| [47] | Experimental | 0.625 | SMF (20) | 10 | OOK/BPSK | NO | NO |
| [48] | Experimental | 1.25 | SMF (25) | 45 | OOK | NO | NO |
| [49] | Experimental | 3.125 | SMF (25) + IDF (25) | 310 | OOK | NO | NO |
| [50] | Analytical Experimental | 0.625 | SMF (20) | 1,5,10,20 | OOK, BPSK, PSM, PAM | NO | NO |
| [52] | Experimental | 1.0 | SMF (20) | 5 | OOK, BPSK | NO | NO |

SMF: Single-Mode fiber; GI-MMF: Graded Index Multimode Fiber; NZDSF: Nonzero dispersion shifted fiber; IDF: inverse dispersion fiber; SOA: Semiconductor Optical Amplifier; OOK: On-Off Keying; PPM: Pulse Position Modulation; BPSK: Binary Phase-Shift Keying; PAM: Pulse Amplitude Modulation; PSM: Pulse Shape Modulation; Q-BOPPM: Quaternary Biorthogonal PPM; FEC: Forward Error Correction Code; CC: Convolutional Codes; LPDC: Low-Density Parity-Check Code.

The rest of the paper is structured as follows. Section 2 presents the methodology, system model and experimental setup of the optical-wireless SLF IR-UWBoF system in-

troduced in this work. The Results and Discussion are presented in Section 3. Finally, conclusions are provided in Section 4.

## 2. Materials and Methods

Several optical implementations for transmitting UWB signals over optical fiber links have been reported in the literature [32–52]. Nevertheless, Impulse Radio (IR)-UWBoF systems that use the Intensity Modulation with Direct Detection (IM/DD) technique have become relevant because they can be directly implemented in optical communication systems (OCS) and widely deployed [53]. Furthermore, recently IR-UWBoF-IM/DD systems have been successfully integrated into Wavelength Division Multiplexing (WDM)-Passive Optical Networks (PONs), demonstrating their applicability for next-generation optical networks [33,38,41–43,47,51]. Considering the latter, the system proposed in this work uses a Radio over Fiber (RoF) architecture based on the IM/DD technique to transmit optical UWB signals [54].

### 2.1. Experimental Methodology

Prior to introducing the system model proposed in this work, this section provides a brief overview of the methodology followed when performing the experiments. As shown in Figure 1, the IR-UWBoF testbed was implemented by first generating an electrical IR-UWB signal by means of MATLAB® and the Arbitrary Waveform Generator (AWG), and then performing electrical to optical conversion. This approach allowed us to implement different IR-UWB modulation schemes in a consistent way, as possible variations caused by the physical implementation of particular pulse generators and modulators were minimized (e.g., variations caused by tolerances of electronic components). Once the electrical IR-UWB signal was generated, it was transmitted first through the optical link and then through the wireless link. The same optical and electronic components were used in the testbed for all performed experiments. This way, we are able to perform a fair comparison between PSDs generated by different modulation schemes used in IR-UWBoF systems, as all IR-UWB signals in the implemented testbed were generated and transmitted using the same hardware.

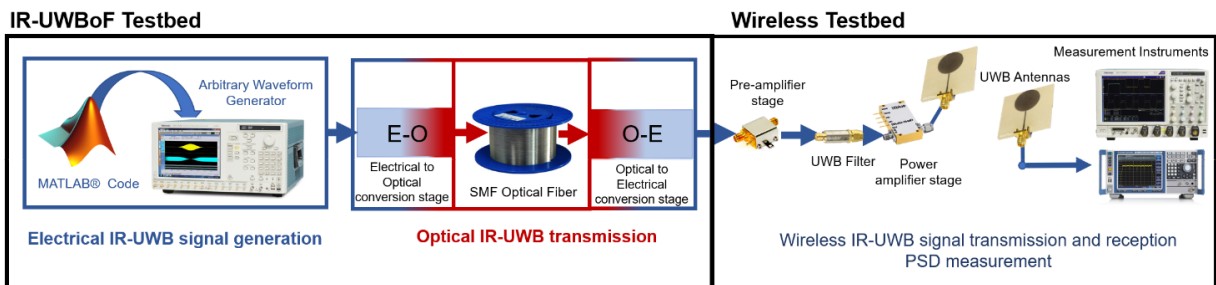

**Figure 1.** Optical-Wireless IR-UWBoF testbed.

In the following subsection, we provide a detailed description of the optical-wireless SLF IR-UWBoF system model introduced in this work.

### 2.2. System Model

The system model proposed in this paper is presented in Figure 2. It consists of an IR-UWB transmitter and an IM/DD RoF architecture. This optical configuration consists of a Continuous Wave (CW) Laser, an External Intensity Modulator (EIM), several kilometers of single-mode fiber, and a PIN-type photodetector. The CW and EIM are used to carry out an electrical to optical conversion process (E/O), and the PIN photodetector is used as an optical to electrical (O/E) converter. Once the electrical signal is obtained at the IM/DD RoF architecture output, this signal is sent to a preamplification stage to compensate for power losses originating from electrical to optical and optical to electrical domain conversions

and optical fiber attenuation. Then, the UWB signal is filtered and transmitted through a wireless channel by using a power amplifier connected to a commercial UWB antenna. The wireless signal reaches an IR-UWB receiver, where the received signal is demodulated and decoded to obtain the transmitted information. The mathematical model of the previously mentioned elements is explained next.

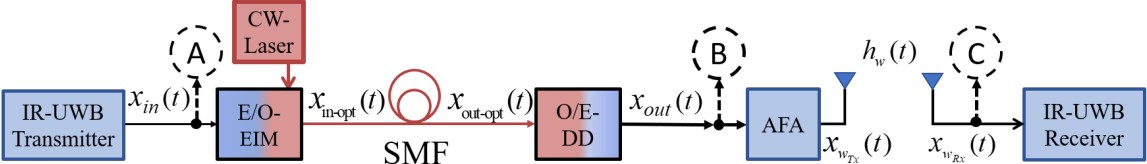

**Figure 2.** Proposed optical-wireless spectral line-free (SLF) IR-UWBoF system; E/O-IM: Electrical to Optical Intensity Modulation; SMF: Single Mode Fiber; O/E-DD: Optical to Electrical Direct Detection; AFA: Amplification, Filtering, Amplification.

### 2.2.1. IR-UWB Transmitter Model

The system model for the IR-UWB transmitter is shown in Figure 3. It is assumed that the binary data source (BDS) generates a data stream, $y_l$, consisting of independent identically distributed (iid) bits with the following probability mass function (pmf), [40]:

$$P[y_l = 0] = p_0, \ P[y_l = 1] = 1 - p_0 = p_1 \tag{1}$$

where $0 \leq p_0 \leq 1$. It is important to mention that $y_l$ is considered to be a perfectly random binary data stream when $p_0 = 1/2$. However, as indicated in [50], this condition is hard to achieve in practical systems.

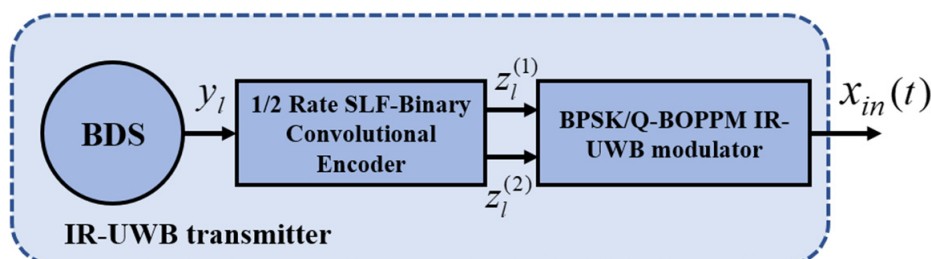

**Figure 3.** Spectral Line-Free (SLF) IR-UWB transmitter diagram block. BDS: Binary Data Source; SLF: Spectral Line-Free; BPSK: Binary Phase Shift Keying; Q-BOPPM: Quaternary Biorthogonal PPM.

The binary data stream, $y_l$, is fed to a rate $1/2$ spectral line-free (SLF) convolutional encoder (CE) such as those proposed in [25]. Thus, for the $l$-th information bit, $y_l$, two binary outputs, $z_l^{(1)}$ and $z_l^{(2)}$, are generated by the convolutional encoder. The feedforward and feedback polynomials of the encoder are $(27,31)_8$ and $(23)_8$, respectively. Figure 4 shows the diagram of the SLF-CE used in this work. Equation (2) presents the mathematical model for the SLF convolutionally coded signal at the output of the BPSK IR-UWB modulator, [40]:

$$x_{in}(t) = \sum_l \left\{ (2z_l^{(1)} - 1)w(t - (2l - 1)T_r) + (2z_l^{(2)} - 1)w(t - (2l)T_r) \right\} \tag{2}$$

For SLF convolutionally coded Q-BOPPM IR-UWB signals, the mathematical expression at the modulator output can be represented as shown in Equation (3), [40]:

$$x_{in}(t) = \sum_l (2z_l^{(1)} - 1)w(t - lT_r - z_l^{(2)}T_\beta) \tag{3}$$

For both, Equation (2) and Equation (3), $l$ is the index of the binary data stream, $y_l$; $x_{in}(t)$ is the IR-UWB transmitted signal; $w(t)$ represents the pulse shape used; $z_l^{(1)}$ and $z_l^{(2)}$ are the SLF-CE outputs; $T_r$ is the mean repetition time between pulses; and $T_\beta$ is the PPM modulation index.

Note that when our proposed system uses a BPSK modulator, two UWB pulses are generated to transmit the equivalent of one information bit. In contrast, when using Q-BOPPM, just one UWB pulse is required to transmit one information bit. Thus, the SLF convolutionally coded BPSK IR-UWBoF system only provides one-half of the data rate offered by the SLF convolutionally coded Q-BOPPM IR-UWB system. As demonstrated in [25], the statistics of the outputs, $z_l^{(1)}$ and $z_l^{(2)}$, are such that when used to drive the BPSK or Q-BOPPM IR-UWB modulators, the generated PSDs will not show any spectral lines, even when the binary data stream, $y_l$, has a pmf with $p_0 \neq 1/2$. However, these capabilities have not been fully demonstrated in UWB transmissions over an optical fiber link concatenated with a wireless channel transmission.

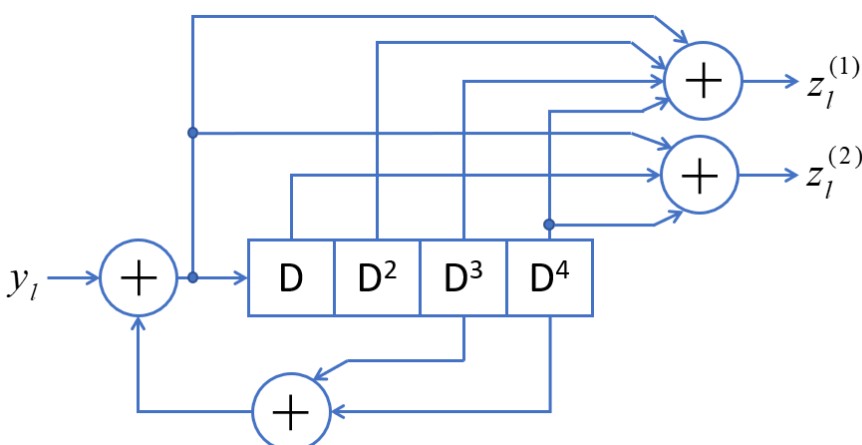

**Figure 4.** Block diagram of 1/2 Rate Spectral Line-Free Convolutional Encoder with feedforward and feedback polynomials $(27,31)_8$ and $(23)_8$, respectively.

2.2.2. Optical Transmitter Model

The intensity modulation with direct detection (IM/DD) radio over fiber (RoF) architecture proposed in this work can be divided into three blocks: electrical to optical (E/O) conversion module, optical channel, and optical to electrical (O/E) conversion module. The mathematical model of the E/O conversion module considers the modulation of a continuous wave (CW) Laser by using an External Intensity Modulator (EIM) based on a balanced arms Mach Zehnder Modulator (MZM). The optical power at the output of this E/O-EIM module is given in Equation (4) as follows, [55],

$$x_{in-opt}(t) = \frac{P_{out-laser}}{2}\left\{1 \pm \sin\left(\frac{\pi V_{RF}(t)}{2V_\pi}\right)\right\} \tag{4}$$

where $x_{in-opt}(t)$ is the optical signal generated by the E/O-EIM module; $V_{RF}(t)$ is the electrical signal provided by the IR-UWB transmitter; $P_{out-laser}(t)$ is the optical power of a CW Laser; and $V_\pi(t)$ is the half-wave voltage of the MZM transfer function. This $V_\pi(t)$ induces a 180° phase change of the optical beam traveling in one arm of MZM [54–56]. It is well known that a sine transfer function can present high or low nonlinearity dependent on the operation voltage. Specifically, bias voltages determine the degree of nonlinearity or linearity of the MZM transfer function. For Microwave Photonics Modulations, biasing an MZM in its linear region such as quadrature (QUAD) bias points $(-V_{\pi/2}, +V_{\pi/2})$ allows transmission of RF broadband signals with multioctave bandwidth [56]. Therefore, care must be taken to maintain and control the MZM bias point for a specific application. Furthermore, it is important to mention that Equation (4) considers a limitless MZM

bandwidth operating in the QUAD point of its transfer function ($\pm V_{\pi/2}$). Thus, nonlinear effects produced by the electrical to optical process are not considered.

The optical channel is formed by several kilometers of standard single-mode fiber (SMF). This optical fiber is usually modeled as shown in Equation (5), [55]:

$$H_{SMF}(f) = \exp\left(j\frac{\lambda^2 D}{c}f^2 L\right) \times \exp\left(-\frac{\alpha_{opt-att}}{2}L\right) \tag{5}$$

where $D$ is the chromatic dispersion parameter; $L$ is the length of the SMF; $\lambda$ is the wavelength of the optical carrier; $c$ is the speed of light in a vacuum; and $\alpha_{opt-att}$ is the attenuation coefficient of the optical fiber. The signal at the optical channel output, $x_{out-opt}(t)$, can be represented by Equation (6), [40]:

$$x_{out-opt}(t) = x_{in-opt}(t) * h_{SMF}(t) \tag{6}$$

where $h_{SMF}(t)$ is the time response of SMF and "$*$" is the convolution operator. It is worth mentioning that the nonlinear effects of the waveguide are neglected in this fiber model.

Finally, the electrical signal at the optical-to-electric conversion module output, $x_{out}$, can be modeled by a PIN-type photodetector without frequency limitations. Equation (7) mathematically models this stage as follows:

$$x_{out}(t) = i(t) = \Re \times x_{out-opt}(t) \tag{7}$$

where $\Re$ is the responsivity of a typical PIN photodetector.

### 2.2.3. Wireless Transmission Model

The EIM/DD RoF architecture provides an attenuated electrical signal due to power losses in conversion modules and SMF transmission. Therefore, this signal must be preamplified, filtered and amplified using a power amplifier before being injected into a UWB antenna. Thus, the wirelessly transmitted UWB signal can be mathematically modelled as follows:

$$x_{wTx}(t) = x_{out}(t) * afa(t) \tag{8}$$

where $x_{wTx}(t)$ is the transmitted wireless UWB signal; $x_{out}(t)$ is the electrical signal given by the RoF architecture; and $afa(t)$ are time responses of electrical modules such as amplifiers, high pass filters and UWB antenna. On the other hand, the received wireless UWB signal, $x_{wRx}(t)$, can be modelled as provided in Equation (9):

$$x_{wRx}(t) = x_{wTx}(t) * h_w(t) \tag{9}$$

where $h_w(t)$ is the time response of a line of sight (LOS) wireless channel.

### 2.3. Optical-Wireless SLF IR-UWBoF Testbed Implementation

The setup diagram of the proposed SLF IR-UWBoF-EIM/DD system and the testbed implementation are shown in Figure 5; Figure 6, respectively. Next, we provide a description of the hardware used to deploy the testbed.

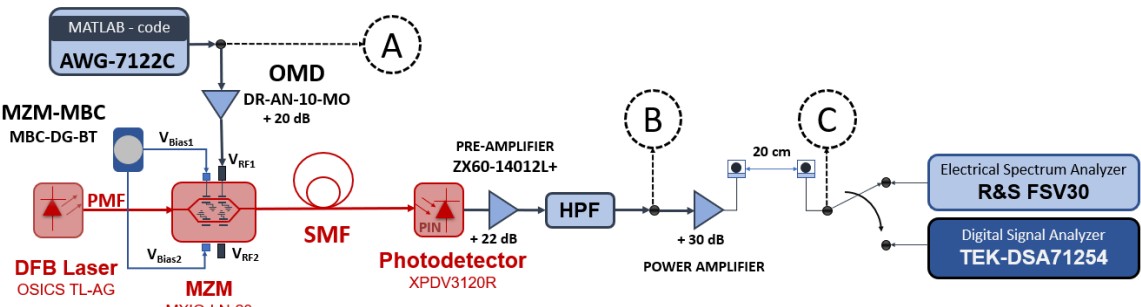

**Figure 5.** Setup diagram of the optical-wireless SLF IR-UWBoF IM/DD system. SLF: Spectral Line-Free; UWBoF: Ultra-Wideband over fiber; IM: Intensity Modulation; DD: Direct Detection; SMF: Single Mode Fiber; PMF: Polarization-maintaining optical fiber; DFB: Distributed-Feedback Laser; MZM: Mach Zehnder Modulator; MBC: Modulator Bias Controller; OMD: Optical Modulator Driver; HPF: High Pass Filter.

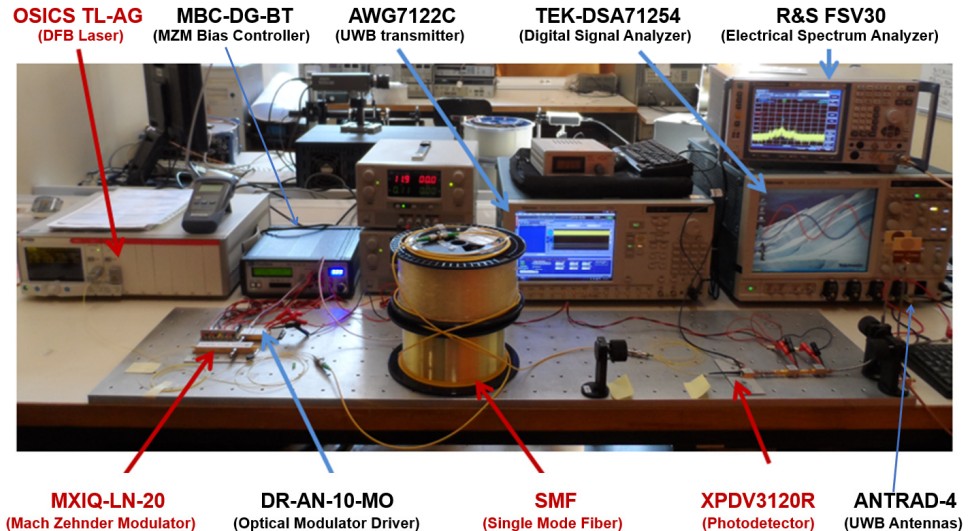

**Figure 6.** Experimental implementation of the optical-wireless SLF IR-UWBoF-EIM/DD system.

### 2.3.1. Electrical IR-UWB Signal Generation

First, the IR-UWB signals with different modulation schemes were generated in MATLAB®. Next, they were loaded into a 9.6 GHz bandwidth arbitrary waveform generator (AWG-7122C from Tektronix). At the output of the AWG (see Test Point A in Figure 5), a 12 GHz electrical modulator driver was connected. This driver was used to match the AWG and the electro-optic modulator impedance.

### 2.3.2. Optical IR-UWB Transmission Implementation

The electrically generated IR-UWB signals modulate the output beam of a 1550 nm Continuous Wave (CW) Distributed Feedback (DFB) laser using an external Dual-parallel Mach Zehnder Modulator (DP-MZM) with 20 GHz electro-optic bandwidth. The optical links were SMF-28 reels of 20 km and 30 km. The IR-UWB signals were electrically recovered at the optical fiber output through a UWB photoreceiver. Then the electrical signals were fed to the wireless testbed implementation.

### 2.3.3. Wireless Testbed Implementation

To compensate for power losses originating from SMF transmission, electrical to optical and optical to electrical conversion processes, a 14 GHz bandwidth amplifier with 22 dB of gain was connected at the photoreceiver output. The amplified signal was then filtered using a UWB high pass filter (HPF).

The signal delivered by the UWB HPF was amplified using a 30 dB gain power amplifier and transmitted through UWB antennas. In our experimental evaluations, the transmit and receive antennas were separated by 20 cm. This separation was enough to evaluate the spectral line suppression capabilities of the proposed system.

The hardware used for wireless transmission consisted of a 12 GHz amplifier with 30 dB gain, and two low-complexity monopole UWB antennas [57]. The Scattering parameters, $S_{11}$ and $S_{21}$, of these electronic components were obtained by using a Vector Network Analyzer (VNA). For electrical amplifiers (with port 1 as the input and port 2 as the output), $S_{11}$ is the input port voltage reflection coefficient describing the level of input-impedance matching, while $S_{21}$ is the forward voltage gain and describes the frequency response. On the other hand, $S_{21}$ for UWB antennas port 1 (input) and port 2 (output) are the transmitter and receiver antennas, respectively. The antenna parameters also include the propagation impairments of the wireless channel, which in the experimental setup is set to be a Line of Sight (LOS) channel to obtain a frequency response reference.

Figure 7 shows the $S_{11}$ and $S_{21}$ scattering parameters of the electrical power amplifier. Figure 7a.1 presents the magnitude of $S_{11}$. It is of particular interest that in the frequency interval from 50 MHz to 7.8 GHz, the magnitude value is below −13 dB. It is well known that when $S_{11}$ is smaller than −13 dB, the impact from the reflections on the transmitted signal is negligible. On the other hand, the magnitude of the transmission parameter, $S_{21}$, is presented in Figure 7a.2. This magnitude plot shows variations below 3 dB in the measured frequency band. Furthermore, it can be seen in Figure 7b.1 and 7b.2 that the power amplifier has a linear phase for both parameters. Therefore, signal distortion is not expected at the output of this electronic device.

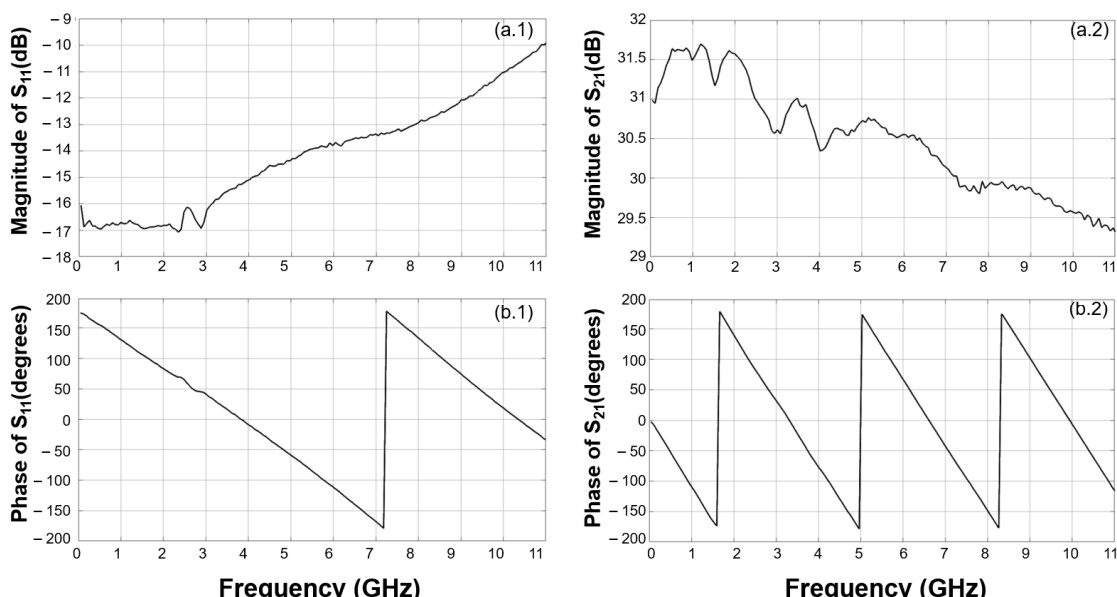

**Figure 7.** Magnitude and phase of $S_{11}$ and $S_{21}$ parameters of the 12 GHz amplifier with 30 dB of gain used in the experimental setup: (**a.1**) $S_{11}$ magnitude, (**a.2**) $S_{21}$ magnitude, (**b.1**) $S_{11}$ phase and (**b.2**) $S_{21}$ phase.

Figure 8 shows the $S_{11}$ parameter (magnitude and phase) of low-cost monopole commercial UWB antennas used in our setup. Figure 8a shows the $S_{11}$ magnitude plot, which is below −13 dB in the frequency interval from 3.3 GHz to 9 GHz. However, in the same interval, there are nonlinearities in the phase (see Figure 8b). Therefore, signal distortion would be expected when UWB signals are transmitted by using these antennas.

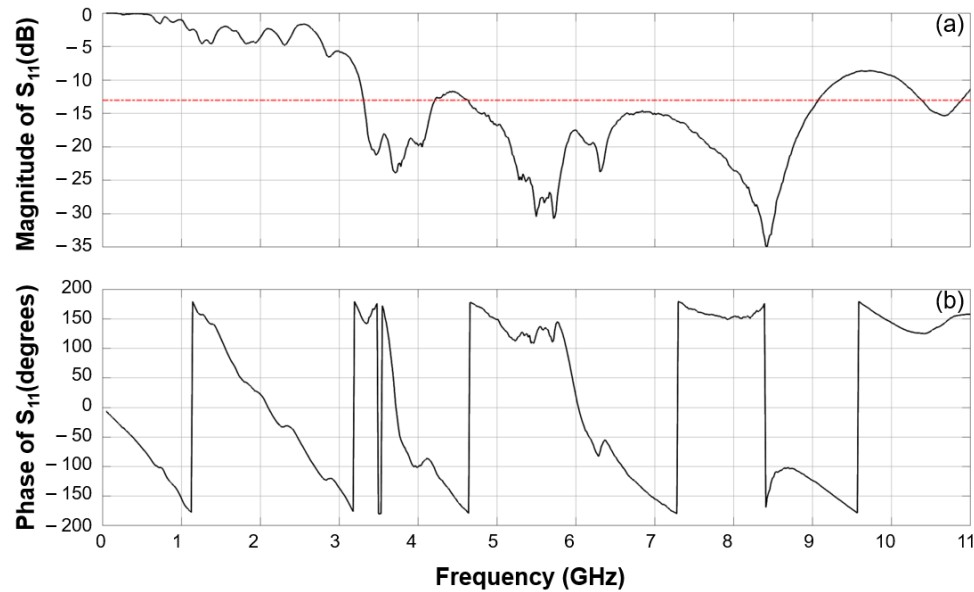

**Figure 8.** Magnitude and phase of $S_{11}$ parameter of the Low-Cost monopole UWB antenna used in the experimental setup:(**a**) $S_{11}$ magnitude, the red plot indicates the $-13$ dB threshold, values smaller than this value are preferred. (**b**) $S_{11}$ phase.

The above is also supported in the UWB antenna datasheet, where the manufacturer provides measurement plots showing that radiation pattern shape depends on frequency and antenna position (see reference 4 from [57]). For example, signals transmitted at 2.5 GHz and 4 GHz exhibit a classic toroidal radiation pattern. On the other hand, the antenna radiation pattern has a butterfly wing shape when 6 GHz or 8 GHz signals are transmitted. It is worth mentioning that the proposed SLF IR-UWBoF-EIM/DD system generates UWB signals that occupy large bandwidths between 3.1 GHz and 9 GHz. Therefore, knowing the frequency response of the wireless transmission setup link would be convenient for designing and testing our proposed system with an aggressive antenna setup configuration in the LOS scenario. Figure 9 presents the UWB positions considered in our tests: front-to-front, back-to-back, side-to-side and front-to-side.

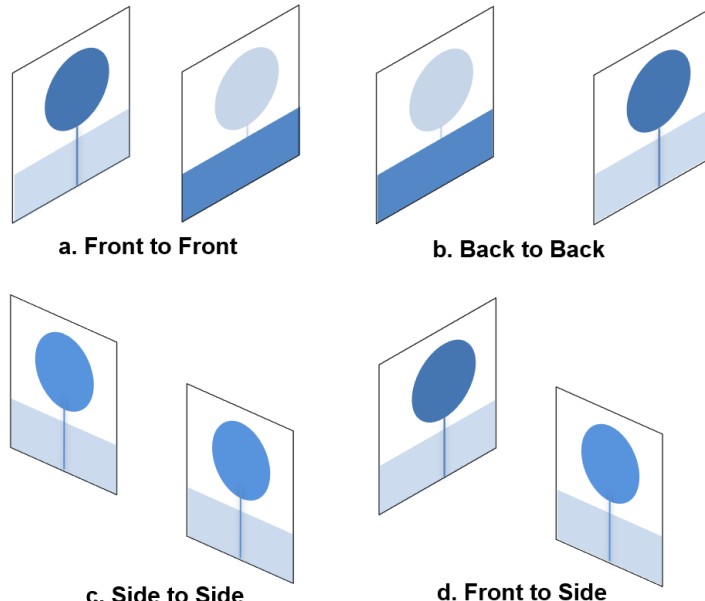

**Figure 9.** Setup of UWB antennas positions in wireless link.

The $S_{21}$ (magnitude and phase) parameter of the Line of Sight (LOS) 20 cm wireless link is presented in Figure 10. Figure 10a–d show the normalized magnitude for front to front, back-to-back, side-to-side, and front-to-side UWB antenna setups, respectively.

Analyzing the plots, we find that the front-to-front and back-to-back configurations provide a reduced bandwidth with notorious phase distortion. This bandwidth reduction and nonlinearities in the phase could have a significant impact on the UWB signals provided by the optical part.

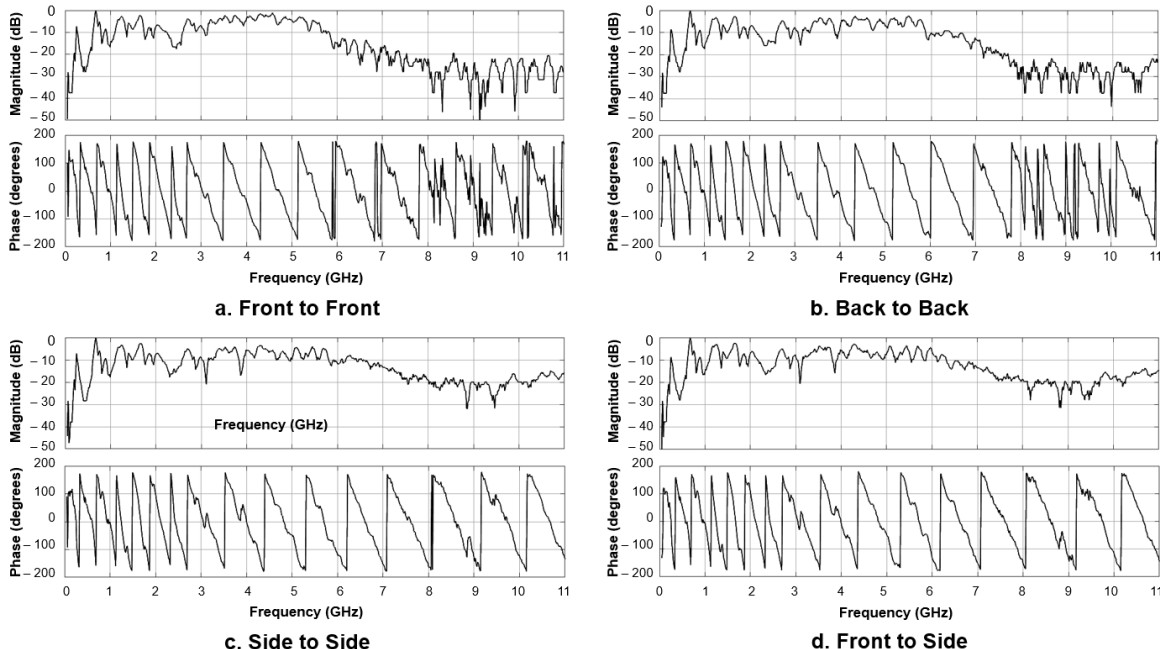

**Figure 10. $S_{21}$** magnitude and phase of the LOS Wireless Link with different antenna positions: (**a**) Front-to-front, (**b**) back-to-back, (**c**) side-to-side and (**d**) front-to-side.

### 2.3.4. PSD Measurement Hardware

To measure the PSD of the IR-UWB signals transmitted through the concatenated optical-wireless links, the output of the receiving antenna (Test Point C in Figure 5) was connected to a 13 GHz real-time digital signal analyzer with 50 GSa/s and a 30 GHz electrical spectrum analyzer (ESA). The ESA resolution bandwidth (RBW) was set to 1 MHz to meet the criteria in [58]. Both instruments were used for digitizing and storing time and frequency domain IR-UWB signals. Using this setup, we were able to obtain the PSD plots introduced in Section 3.

It is worth mentioning that additional instruments were used in our setup to ensure stable operation during extended work periods. For example, the bias drift phenomena, caused by thermal effects, cause the transfer function of the MZM to drift over time and change the optimum bias point setting, [56]. Therefore, a Mach–Zehnder Modulator Bias Controller (MBC) was used to maintain the MZM bias voltages at the QUAD operational point of its transfer function. As previously mentioned, biasing an MZM in its QUAD bias point enables an electrical-to-optical conversion process with minimum nonlinear effects (e.g., signal distortion) on the output signal.

### 3. Results and Discussion

This section introduces power spectral density (PSD) measurements obtained from the proposed SLF convolutionally coded BPSK/Q-BOPPM IR-UWBoF-IM/DD systems. PSD measurements of traditional noncoded BPSK IR-UWBoF systems are also reported for comparative purposes.

The signal parameters used to evaluate SLF convolutionally coded BPSK/Q-BOPPM IR-UWBoF-IM/DD systems are as follows: $T_r = 1$ ns, $T_\beta = 0.5$ ns, and $w(t) = 5th$ derivative of a Gaussian pulse with pulse duration $T_w \approx 0.5$ ns. For each PSD measurement, a data stream, $y_l$, consisting of 10,000 bits, was generated in MATLAB® using the method reported in [31].

Measurements were made considering two probability distributions for the data stream, one with $p_0 = 1/5$ and another with $p_0 = 2/5$. For the case of the noncoded IR-UWB systems, the data stream, $y_l$, is used to drive the BPSK UWB modulator. It is worth mentioning that a maximum data rate of 1 Gbps can be achieved with the SLF convolutionally coded Q-BOPPM IR-UWB-IM/DD system, whereas a maximum data rate of 500 Mbps can be achieved with the SLF convolutionally coded BPSK IR-UWB system. Once coded and noncoded IR-UWB signals were generated in MATLAB®, they were loaded into the AWG7122C to generate the corresponding electrical signal. It is important to note that the amplitude of the base pulse and the pulse generation rate was the same for all experiments, i.e., all transmitted signals had the same transmit power.

Figure 11 shows measured PSDs of noncoded and SLF convolutionally coded IR-UWB signals that follow Equations (2) and (3). These signals were measured at Test Point B (TPB) in Figure 5. The light blue plots in Figure 11 correspond to the measured PSD of signals where the data stream has a pmf with $p_0 = 1/5$. The dark blue plots correspond to the measured PSD of signals where the data stream has a pmf with $p_0 = 2/5$. The dotted red line indicates the FCC UWB spectral mask limits for indoor communications. As shown in Figure 11a.1–a.3, the noncoded BPSK signals (labelled NC-BPSK in the figure) exhibit solid spectral lines (SSLs) in the PSD. These spectral lines are observed because the data stream used to generate the UWB signal is not perfectly random ($p_0 \neq 1/2$), as is the case in most practical cases.

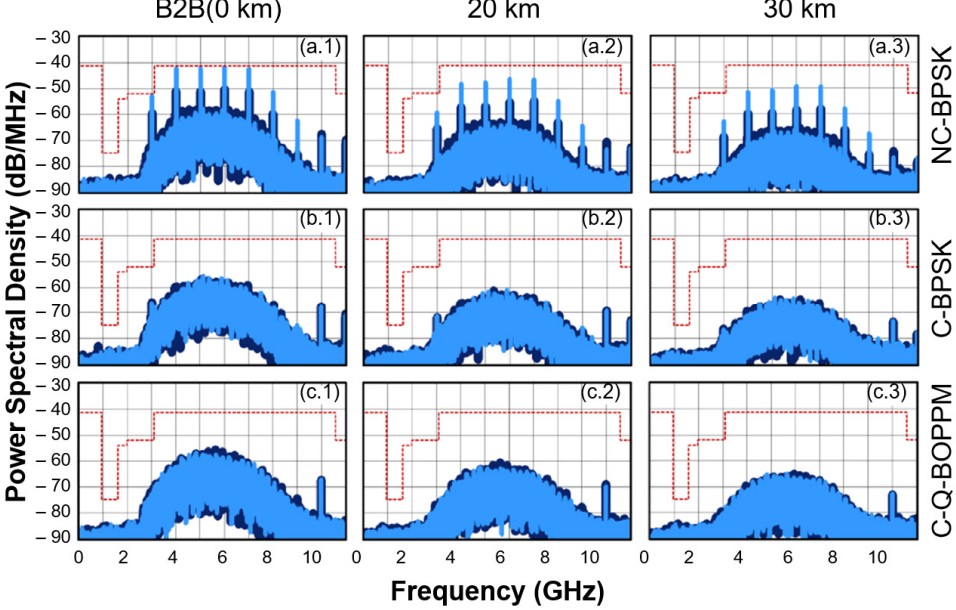

**Figure 11.** Measured PSDs of signals generated by: (**a**) the noncoded -BPSK IR-UWBoF-EIM/DD system (first row labelled as NC-BPSK—plots **a.1**, **a.2**, and **a.3**); (**b**) the SLF convolutionally coded BPSK IR-UWBoF-EIM/DD system (second row labelled as C-BPSK—plots **b.1**, **b.2**, and **b.3**); and (**c**) the SLF convolutionally coded Q-BOPPM IR-UWBoF-EIM/DD system (third row labelled as C-Q-BOPPM—plots **c.1**, **c.2**, and **c.3**). The PSDs were obtained at Test Point B, pointed out in Figure 5, after 0 km (first column—plots **a.1**, **b.1**, and **c.1**), 20 km (second column—plots **a.2**, **b.2**, and **c.2**) and 30 km (third column—plots **a.3**, **b.3**, and **c.3**) of SMF.

In contrast, PSDs of SLF convolutionally coded IR-UWB signals (labelled as C-BPSK and C-Q-BOPMM in the figure) shown in Figure 11b.1–b.3,c.1–c.3, do not show solid spec-

tral lines. These results agree with those reported in [40], where a spectral line suppression condition is fulfilled over 30 km SMF transmission. It is important to mention that the spectral line observed near 10 GHz comes from nonlinearities of the AWG. This spectral line could be attenuated using a notch filter.

The proposed SLF convolutionally coded IR-UWBoF-IM/DD system was also evaluated over a concatenated channel formed by 20 km and 30 km SMF reels and a 20 cm wireless link. The PSDs measured at the receiver stage, corresponding to test point C in Figure 5, are shown in Figure 12. We can observe that for the noncoded BPSK system, the SSLs are kept in the PSDs measured after wireless transmission.

For the convolutionally coded signals (labelled as C-BPSK and C-QBOPPM in Figure 12) the measured PSDs do not show significant spectral lines after wireless transmission. These results demonstrate that the SLF condition is held by our proposed SLF convolutionally coded IR-UWBoF-IM/DD scheme in concatenated optical-wireless channels. Furthermore, it can be seen in Figure 12 that for the coded systems, the transmit power can be increased by at least 8 dB without exceeding the spectral mask limits, which would further improve the bit error rate (BER) performance. By comparing Figures 11 and 12, it can be observed that a bandwidth reduction in the measured PSDs occurs after wireless transmission. This is caused by the limited frequency response of the commercial high pass filters and UWB antennas used in the experimental setup. Although these electronic devices do not generate spectral lines, the UWB pulse broadening in the time domain could reduce the maximum data rate achievable by the system.

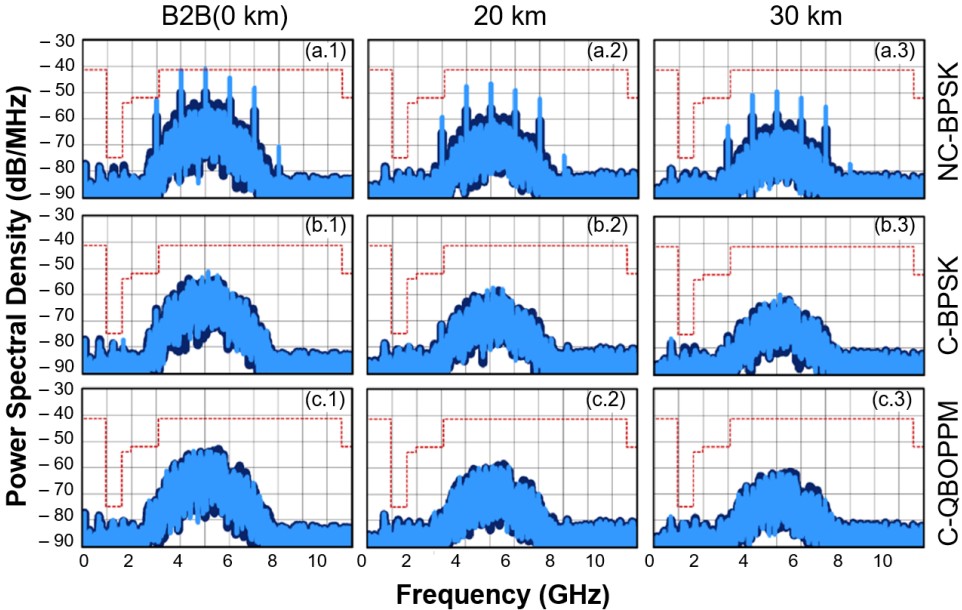

**Figure 12.** Measured PSDs of signals generated by: (**a**) the noncoded -BPSK IR-UWBoF-EIM/DD system (first row labelled as NC-BPSK—plots **a.1**, **a.2** and **a.3**); (**b**) the SLF convolutionally coded BPSK IR-UWBoF-EIM/DD system (second row labelled as C-BPSK—plots **b.1**, **b.2** and **b.3**); and (**c**) the SLF convolutionally coded Q-BOPPM IR-UWBoF-EIM/DD system (third row labelled as C-Q-BOPPM—plots **c.1**, **c.2** and **c.3**). The PSDs were obtained at Test Point C, pointed out in Figure 5, after 0 km (first column—plots **a.1**, **b.1** and **c.1**), 20 km (second column—plots **a.2**, **b.2** and **c.2**) and 30 km (third column—plots **a.3**, **b.3** and **c.3**) of SMF and wireless transmission.

## 4. Conclusions

The spectral line suppression capabilities of the proposed SLF convolutionally coded BPSK/Q-BOPPM IR-UWBoF-EIM/DD systems were thoroughly evaluated over optical and wireless channels. The results reported in Section 3 confirm that the proposed system can

provide spectral line-free (SLF) PSDs over concatenated channels, even when nonuniformly distributed binary data streams are transmitted.

Note in Figure 12 that the PSD maximum for the proposed SLF convolutionally coded BPSK/Q-BOPPM IR-UWBoF-EIM/DD systems is at least 10 dB below the PSD maximum observed for the noncoded system. Therefore, the transmit power of the system proposed in this work could be significantly increased while still maintaining spectral mask compliance. This would improve the transmission range and BER performance of the proposed system compared with the conventional noncoded system, even before considering any possible coding gain provided by the convolutional code. Detailed analysis of BER performance improvements achieved with the proposed SLF convolutionally coded BPSK/Q-BOPPM IR-UWBoF-EIM/DD systems compared to traditional BPSK/Q-BOPPM IR-UWBoF implementations will be addressed in future work.

Finally, we would like to mention that the reported systems could be used to interconnect IR-UWB WSN deployments separated by several tens of meters, even kilometers. They could also be integrated into next generation networks such as WDM-PON systems to offer wireless services within the Internet of Things and smart cities paradigms.

**Author Contributions:** Conceptualization, A.-E.P.-R., S.V.-R. and A.G.-M.; methodology, A.-E.P.-R., S.V.-R., A.G.-M. and C.L.; software, A.-E.P.-R. and C.L.; validation, A.-E.P.-R., S.V.-R. and C.L.; formal analysis, A.-E.P.-R. and S.V.-R.; investigation, A.-E.P.-R., S.V.-R., A.G.-M. and C.L.; resources, S.V.-R. and C.L.; data curation, A.-E.P.-R. and C.L.; writing—original draft preparation, A.-E.P.-R.; writing—review and editing, A.-E.P.-R., S.V.-R. and A.G.-M.; visualization, A.-E.P.-R. and A.G.-M.; supervision, S.V.-R. and C.L.; project administration, S.V.-R.; funding acquisition, S.V.-R., A.G.-M. and C.L. All authors have read and agreed to the published version of the manuscript.

**Funding:** This research was partially funded by CONACYT México under Basic Scientific Research grant number 2016-01-285276 and SEP-CONACYT-ECOS Nord-ANUIES project with grant number M09P03. The APC was funded by CONACYT México under Basic Scientific Research grant number 2016-01-285276.

**Institutional Review Board Statement:** Not applicable.

**Informed Consent Statement:** Not applicable.

**Data Availability Statement:** Not applicable.

**Acknowledgments:** The authors would like to thank SAMOVAR laboratory for allowing us to use the AWG7122C and DSA71254 equipment to perform the experiments reported in this paper.

**Conflicts of Interest:** The authors declare no conflict of interest. The funders had no role in the design of the study; in the collection, analyses, or interpretation of data; in the writing of the manuscript; or in the decision to publish the results.

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
