# Peer review of "Design of Optical-Wireless IR-UWBoF Systems with Spectral Line Suppression Capabilities"

_electronics, doi:10.3390/electronics11213496_

Round 1
Reviewer 1 Report
1. Give proper reference to the equations, if they are taken from previously published literature.
2. The quality of some figures is not good. It is suggested to improve the quality of the figures before publication.
3. The English language needs some improvement.
Reviewer 2 Report
Title: Design of optical-wireless IR-UWBoF systems with spectral line suppression capabilities
The authors have carried out the study in a well-organized manner. The presentation of problem, limitations, and methodology of the study are presented in a structured manner. However, the following comments must incorporate for the further improvements
Comments
1. The motivation and contribution of the study must be included at the end of the introduction.
2. The abstract of the study need to be balance with background, problem statement, previous studies limitations on the same problem statement, proposed methodology, numerical findings of the model, novelty and originality of the study
3. The methodology of the proposed study needs to present in pictorial representation for better readability.
4. A comparative analysis of the previous studies and proposed study must be presented for concluding the novelty and originality. In addition to more latest references (2021 and 2022) need to be incorporated in the study.
5. The authors must highlight the scientific numerical findings with its novelty in the conclusion section. Along with this, the future work and applications of the proposed model.
Reviewer 3 Report
The manuscript “Design of optical-wireless IR-UWBoF systems with spectral line suppression capabilities” presents a novel approach regarding a different architecture aiming to eliminate spectral lines from the power spectral density in wireless ultra-wideband systems. The concatenated system of optical fiber and optical wireless communication system supporting Impulse-Radio Ultra-Wide Band transmission proposed here by the authors, complies with the power limits and spectral masks provided by radio spectrum regulatory agencies.
Before submitting the final form of the manuscript for publication, please take into consideration the following suggestions:
SLF acronym mentioned in the abstract does not have an explanation. It can be found on the line 76 only.
The description placed under the Figures (3, 4, and 7) should be revised and more accurate.
For example:
- Figure 3. SLF IR-UWB transmitter diagram block
o Delete the BDS in brackets
o At the end of BPPM replace the semicolon ; with the :
- In Figure 4. SLF IR-UWB transmitter diagram block
o Explanations are not related to the acronyms in the figure above;
o there is no explanation about SMF (?) and a better correlation in terms between Figures 4 and 5 would be more appreciated.
- Figure 7. PSD signals of Noncoded (NC)-BPSK IR-UWBoF-EIM/DD system and spectral line free 279 (SLF)-convolutional encode (CE) BPSK and SLF-CE Q-BOPPM IR-UWBoF-EIM/DD systems.
o Here the test point is C (not B), according to the mentione in the paragraph above (line 274).
Delete lines from 234 to 237: “This section may be divided into subheadings. It should provide a concise and precise description of the experimental results, their interpretation, as well as the experimental conclusions that can be drawn.”
There should be a more extensive presentation about the wireless optical transmission hardware (with their key characteristics) used in the experiment presented in figures 4 and 5.
Maybe the sentence “It is worth mentioning that additional instruments were used in our setup to ensure stable operation during extended work periods” refers to this equipment, even so, they should be mentioned and their key characteristics relevant to the experiments and the results presented, should be visible for the reader.
